# INVERTED ACTIVATIONS: REDUCING MEMORY FOOTPRINT IN NEURAL NETWORK TRAINING

## ABSTRACT

The scaling of neural networks with increasing data and model sizes necessitates the development of more efficient deep learning algorithms. A significant challenge in neural network training is the memory footprint associated with activation tensors, particularly in pointwise nonlinearity layers that traditionally save the entire input tensor for the backward pass, leading to substantial memory consumption. In this paper, we propose a modification to the handling of activation tensors in pointwise nonlinearity layers. Our method involves saving the output tensor instead of the input tensor during the forward pass. Since the subsequent layer typically also saves its input tensor, this approach reduces the total memory required by storing only one tensor between layers instead of two. This optimization is especially beneficial for transformer-based architectures like GPT, BERT, Mistral, and Llama. To enable this approach, we utilize the inverse function of the nonlinearity during the backward pass. As the inverse cannot be computed analytically for most nonlinearities, we construct accurate approximations using simpler functions. Experimental results demonstrate that our method significantly reduces memory usage without affecting training accuracy or computational performance. Our implementation is provided as a drop-in replacement for standard nonlinearity layers in the PyTorch framework, facilitating easy adoption without requiring architectural modifications. The code is available at `https://github.com/removed/for/anonimity`.

## 1 INTRODUCTION

The remarkable scaling of neural networks with increasing data and model sizes has created a continuous demand for training and deploying larger models more efficiently. This necessity drives the ongoing development of deep learning algorithms that are both computation and memory-efficient.

A significant portion of memory consumption during neural network training arises from activation tensors, alongside model parameters and optimizer statistics. Activation tensors are stored during the forward pass to facilitate gradient computations in the backward pass. Even simple and computationally fast operations, such as pointwise nonlinearities, contribute substantially to this memory footprint by storing entire input tensors. In most popular deep learning frameworks, pointwise nonlinearities like GELU Hendrycks & Gimpel (2016) and SiLU Elfwing et al. (2018) save the whole

```python
model = create_model_as_usual()
for layer in model.layers:
    layer.act_fn = InvActGELU()
train_as_usual(model)
```

Listing 1: Pseudocode of hypothetical use of our drop-in replacement for GELU that will use less memory without any speed or quality degradation. Exact memory gains for several popular huggingface[1] models are presented in the following table. Exact model setups are given in appendix A.1.

| Model | Memory Saving |
|---|---|
| **BERT** | $-22.9\%$ |
| **Audio Spectral Transformer** | $-24.0\%$ |
| **ViT** | $-23.8\%$ |
| **CLIP** | $-23.4\%$ |

---

[1]`https://huggingface.co/models`

input tensor for the backward pass. This practice leads to considerable memory usage, especially in transformer-based architectures where such activations are prevalent.

In this work, we propose a modification to the forward and backward passes of element-wise nonlinearity layers. Specifically, we suggest saving **the output tensor instead of the input tensor** during the forward pass. Since the subsequent layer in the computation graph typically also saves its input tensor, our approach ensures that only one tensor is stored between layers rather than two, effectively reducing the memory footprint by nearly **25%** in practice.

For a novel approach in implementing nonlinearity functions to gain widespread adoption, it should ideally meet three key conditions:

1. The method should not introduce any additional computational error, ensuring that model performance remains unaffected.

2. It should not slow down the model, maintaining or improving the speed of forward and backward computations.

3. The approach should be user-friendly, meaning it can serve as a straightforward, drop-in replacement for existing nonlinearity layers without requiring modifications to the architecture or specific tuning of adjacent layers.

For example, highly optimized building blocks like Triton NVIDIA Corporation or CUDA-based GeGLU implementations for models such as Llama and Mistral, or efficiently fused MLP blocks for transformer architectures, can offer significant speed and/or memory advantages, but are applicable to a very narrow range of use cases. In contrast, for general research purposes and R&D, truly versatile layers are highly desirable. In this paper, we demonstrate that our proposed method meets all three of these criteria, making it a promising candidate for broader adoption in deep learning systems.

We implemented this approach as a drop-in replacement for standard nonlinearity layers within the PyTorch Ansel et al. (2024) framework, ensuring seamless integration for researchers and practitioners without the need for architectural modifications. In Listing 1, we illustrate the straightforward integration of our nonlinearity layers into existing pipelines, which requires only the substitution of standard nonlinearities with our InvAct (Inverted Activation) layers, without the need for additional modifications to the architecture or hyperparameters.

## 2 METHOD

**Background Knowledge.** The computation graph during neural network training can be expressed as a series of operations $f_i$, each of which transforms an input tensor $\mathbf{X}^{(i)}$ into an output tensor $\mathbf{X}^{(i+1)}$. This output tensor subsequently serves as the input for the next operation in the computation graph:

$$\ldots \to \mathbf{X}^{(i)} \xrightarrow{f_i} \mathbf{X}^{(i+1)} \to \ldots \to L.$$

Here, $L$ is the final loss that is optimized using gradient descent. Each operation stores some additional information $\mathbf{S}^{(i)}$, referred to as activation tensors, for the backward pass. During the backward pass, these activation tensors are used to compute the gradient of the loss $L$ with respect to all intermediate tensors of the computation:

$$\ldots \leftarrow \frac{\partial L}{\partial \mathbf{X}^{(i)}} \xleftarrow{\text{gradient } f_i, \mathbf{S}^{(i)}} \frac{\partial L}{\partial \mathbf{X}^{(i+1)}} \leftarrow \frac{\partial L}{\partial L}.$$

For pointwise nonlinearity layers, the forward computation $f_i$ is simply the element-wise application of a function $f : \mathbb{R} \to \mathbb{R}$ to the input tensor $\mathbf{X}^{(i)}$:

$$\mathbf{X}^{(i+1)}_{j_1,\ldots,j_k} = f\left(\mathbf{X}^{(i)}_{j_1,\ldots,j_k}\right), \tag{1}$$

while the backward computation is the Hadamard product of the gradient with respect to the output tensor and the derivative of $f$ evaluated at the input tensor:

$$\frac{\partial L}{\partial \mathbf{X}^{(i)}} = \frac{\partial L}{\partial \mathbf{X}^{(i+1)}} \odot f'\left(\mathbf{X}^{(i)}\right). \tag{2}$$

```python
def forward(X):
    S = X < T
    Y = f(X)
    save_for_backward(Y, S)
    return Y

def backward(dY):
    Y, S = saved_for_backward()
    return dY * f'(f⁻¹(Y, S))
```

Listing 2: Pseudocode of inverted activation approach for forward and backward passes of pointwise nonlinearity layer. Boolean array S should be saved in an optimal way (1 bit per element) to achieve memory footprint reduction.

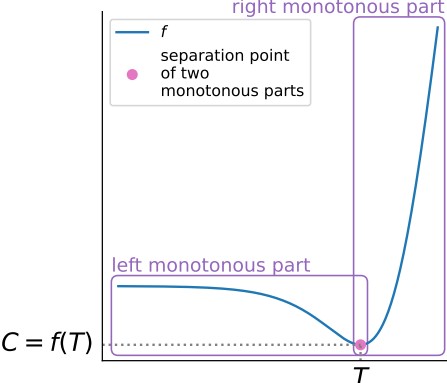

Figure 1: Two monotonous parts of the GELU or SiLU nonlinearity function, each being an invertible function.

Here, $f'\left(\mathbf{X}^{(i)}\right)$ is computed element-wise, similarly to Eq. (1).

Since both Eq. (1) and Eq. (2) are applied independently to each element of the corresponding tensors, we can simplify the notation by letting $x \in \mathbb{R}$ represent an input element of the nonlinearity and $y \in \mathbb{R}$ represent the corresponding output:

$$y = f(x), \quad \frac{\partial L}{\partial x} = \frac{\partial L}{\partial y} f'(x). \tag{3}$$

It is evident that to perform the backward computation in Eq. (2) efficiently, one can store the entire input tensor $\mathbf{X}^{(i)}$ as an activation tensor $\mathbf{S}^{(i)}$. This approach is adopted by most popular deep learning frameworks, such as PyTorch and Jax, for nonlinearity layers like GELU and SiLU, which are widely used in transformer-based neural networks.

**Inverted Activation.** In this work, we propose modifying the forward and backward computations to save the output tensor $\mathbf{X}^{(i+1)}$ instead of the input tensor $\mathbf{X}^{(i)}$. If the subsequent layer after the pointwise nonlinearity also saves its input tensor, the two layers together will only store one tensor instead of two, thereby significantly reducing the overall memory footprint.

The backward computation in Eq. (3) can be rewritten as:

$$\frac{\partial L}{\partial x} = \frac{\partial L}{\partial y} f'(x) = \frac{\partial L}{\partial y} f'\left(f^{-1}(y)\right).$$

Here, $f^{-1}$ is the inverse function of the nonlinearity $f$, which gives our method its name: **Inverted Activations** (or **InvAct** for short).

**Space-efficient Boolean Indicator.** Both GELU and SiLU are non-invertible functions, but they consist of two monotonic parts, each of which is individually invertible. Thus, we can save additional information during the forward pass – a Boolean indicator that specifies whether each element of the input tensor belongs to the first monotonic part or the second part, which helps disambiguate $f^{-1}$:

$$s = \begin{cases} 1 & \text{if } x < T \\ 0 & \text{otherwise} \end{cases} \tag{4}$$

Here, $T$ is the point of separation between the two monotonic halves (see Fig. 1 for a graphical explanation). It is possible to store the tensor $\mathbf{S}$ in a space-efficient Boolean array, where each element requires only 1 bit of memory. In our implementation, we achieve this by storing the Boolean array $\mathbf{S}$ in a tensor $\mathbf{S}_{\text{compressed}}$ with the data type 'uint8', in which $\mathbf{S}[i]$ corresponds to the ($i$ mod 8)-th bit of the ($i \div 8$)-th element:

$$\mathbf{S}[i] = \mathbf{S}_{\text{compressed}}[i \div 8] \gg (i \mod 8).$$

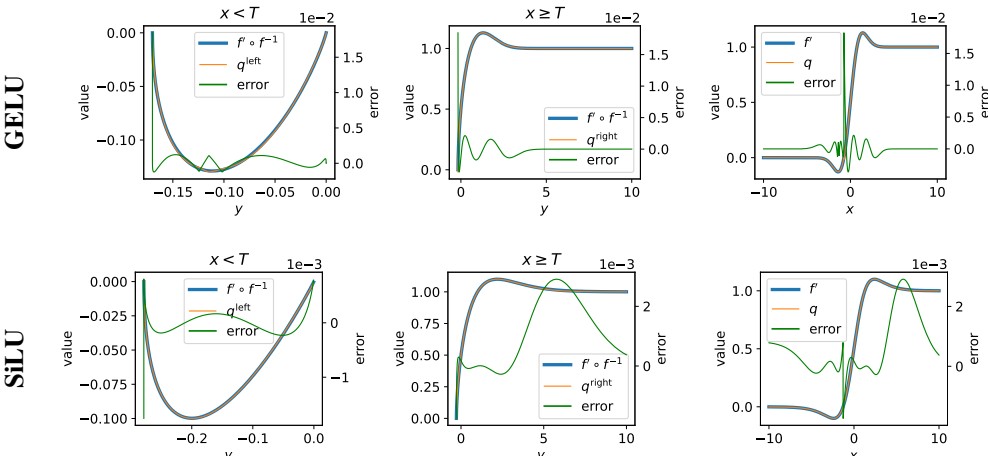

Figure 2: Derivative of the nonlinearity and its approximation in two coordinate systems: plotted as a function of the output variable $y$ (left two columns) and the input variable $x$ (right column). The $y$-based representation highlights how equations Eqs. (5) to (8) approximate the respective regions of $f'(f^{-1}(y))$. The $x$-based representation, in turn, helps assess the approximation quality of the layer baased on how it is used in a neural network context.

Here, $\gg$ denotes the bitwise right shift, and $\&$ denotes the bitwise AND operation. Pseudocode for the resulting pointwise nonlinearity layer can be found in Section 2.1.

**Approximation.** Both GELU and SiLU are complex functions, and to the best of our knowledge, neither their inverses $f^{-1}$ nor $f' \circ f^{-1}$ can be analytically derived. Therefore, we propose to use approximations based on simple base functions, similar to the fast GELU approximation Hendrycks & Gimpel (2016).

There are many decisions to be made when constructing the approximation for $f' \circ f^{-1}$: whether to approximate only $f^{-1}$ and evaluate $f'$ in the standard way, or to approximate $f' \circ f^{-1}$ as a whole; which primitive functions to use; what form the approximation should take; and how to balance complexity and accuracy. In this work, we chose to approximate $f' \circ f^{-1}$ as a whole. The resulting approximation is highly accurate – so much so that the training process using our approximation is indistinguishable from training with the original nonlinearity layer, as we will demonstrate in our experiments in Section 3.

For GELU we use the following approximations:

$$x < T \Rightarrow f'\left(f^{-1}(y)\right) \approx q^{\text{left}}(y) = c_0\sqrt{y + c_1} * (2y + c_2\sqrt{-y})(|c_3 y^2 + |c_4 y + c_5| + c_6| + c_7), \tag{5}$$

$$x \geq T \Rightarrow f'\left(f^{-1}(y)\right) \approx q^{\text{right}}(y) = 1 + \left(c_0 + c_1\sqrt{\tilde{y}} + c_2\tilde{y}\right)e^{c_3(c_4 - \tilde{y})^3}, \text{ where } \tilde{y} = y - f(T). \tag{6}$$

For SiLU we use the following approximations:

$$x < T \Rightarrow f'\left(f^{-1}(y)\right) \approx q^{\text{left}}(y) = \left(c_0 + c_1\sqrt{\tilde{y}} + c_2\tilde{y} + c_3\tilde{y}^2\right) * (1 - y) + y, \tag{7}$$

$$x \geq T \Rightarrow f'\left(f^{-1}(y)\right) \approx q^{\text{right}}(y) = \left(1 + \left(c_0 + c_1\sqrt{\tilde{y}} + c_2\tilde{y}\right)e^{c_3(c_4 - \tilde{y})^3}\right) * (1 - y) + y, \tag{8}$$

$$\text{where } \tilde{y} = y - f(T) \tag{9}$$

The coefficients $c_i$ are precomputed and fine-tuned specifically for each function (GELU or SiLU) and each approximation region ($x < T$ or $x \geq T$). All coefficient values used in this work are provided in appendix A.2. In Fig. 2, we illustrate the functions $f'(f^{-1}(y))$ for $x < T$ and $x \geq T$, along with their respective approximations $q^{\text{left}}$ and $q^{\text{right}}$. The plots also show the approximation errors, defined as $f'(f^{-1}(y)) - q^{\text{left}}(y)$ for $x < T$ and $f'(f^{-1}(y)) - q^{\text{right}}(y)$ for $x \geq T$.

It is important to note that the presented approximations are not necessarily the most optimal or minimal representations of $f'(f^{-1}(y))$. As we are not experts in constructing such approximations, it is possible that simpler, more accurate, and computationally efficient alternatives exist, which could further enhance the performance of InvAct. Nonetheless, our current approximations are well-suited for the specific task at hand, as demonstrated by their effectiveness, accuracy and speed in the experiments detailed in Section 3.

## 2.1 ALTERNATIVES TO BIT-COMPRESSED BOOLEAN INDICATOR

An essential component for inverting nonlinearities such as GELU or SiLU is an indicator function that identifies which monotonic part the original $x$ belongs to. Currently, we store this information in a space-efficient, bit-compressed Boolean tensor, where each Boolean value occupies only 1 bit of memory. In this section, we will discuss two alternative methods for storing this information without requiring even 1 bit per element.

**Sign-bit Inverted Activation.** Both GELU and SiLU functions are bounded, which means there exists some constant $C$ (see Fig. 1) such that for all $x$, $f(x) \geq C$, where $f$ represents either the GELU or SiLU function. This implies that the value $f(x) - C$ is always greater than zero. Instead of storing $f(x)$ for the backward pass, we can store $f(x) - C$ with its sign bit modified according to the indicator function $s$ (see Eq. (4)).

For most popular floating-point formats used in deep learning, such as float32, float16, and bfloat16, the sign bit is the last bit of a floating-point number. This approach allows us to eliminate the need for additional memory to store the 1 bit per element for the indicator function, making the entire nonlinearity layer memory-free in terms of footprint. The modified value $y = f(x)$ does not pose an issue for subsequent computations, as $C$ is a predefined global constant for a given activation function. For example, the computation for an activation function followed by a fully-connected linear layer can be expressed as follows:

```
N   # number of bits in floating point data type
C   # minimum of activation function f
def forward(X, W_linear):
  S = X < T
  Y = f(X) - C
  Y = (Y & (~(1 << N))) | (S << N)   # sets sign bit of tensor Y
                                     # to indicator function S
  save_for_backward(Y)

  return (Y + C) @ W_linear

def backward(dY, W_linear):
  Y = saved_for_backward()
  S = Y & (1 << N)# extracts indicator function from last bit of tensor Y
  Y = Y ^ S        # reverts sign bit of tensor Y back to zero
  dW = (Y + C).T @ dY
  dX = dY * f`(f^{-1}(Y + C, S))
  return dY, dW
```

Here, the forward pass takes the input tensor $X$ and the matrix $W_{\text{linear}}$ of the fully-connected linear layer and returns $f(X) \cdot W_{\text{linear}}$.

Unfortunately, to the best of our knowledge, it is not possible to implement such a layer in PyTorch that works independently of what comes after the nonlinearity function, meaning that you would have to write a custom implementation for each new architectural block. As a result, the sign-bit approach does not satisfy condition 3 outlined in the Introduction (see Section 1), and thus we do not consider it a viable candidate for broad adoption.

**Precision Bit Inverted Activation.** The second approach to store the indicator function $s$ (see Eq. (4)) is to use the lowest precision bit of a floating-point number (for most popular floating-point data types, this is the first bit in a base-2 representation). During the forward pass, we compute $y = f(x)$ and modify this bit of $y$ to be equal to $s$ (see Eq. (4)).

Compared to the previous approach, this method does not violate condition 3, as it allows for a convenient drop-in replacement for classical activation functions on all modern deep learning frameworks. However, it is more challenging to evaluate the accuracy of this approach. Both the standard inverted activation and the sign-bit inverted activation perform the forward pass exactly and introduce only a minimal approximation error during the backward pass. This method, on the other hand, sacrifices one precision bit for the pointwise nonlinearity computation during the forward pass. Moreover, with the ongoing trend of reducing floating-point precision, loss of accuracy may become larger in the future. For this reason, we do not consider it a viable candidate for broad adoption at this time, as it requires extensive experimentation to validate its accuracy. However, we still regard it as an interesting approach that may find applications in the future.

This is a pseudocode for the forward and backward passes of this approach:

```
1  def forward(X):
2      S = X < T
3      Y = f(X)
4      Y = (Y & (~1)) | S
5      save_for_backward(Y)
6
7      return Y
8
9  def backward(dY):
10     Y = saved_for_backward()
11     S = Y & 1 # extracts indicator fucntion from last bit of tensor Y
12     return dY * f`(f⁻¹(Y, S))
```

## 3 EXPERIMENTS

The validation of our method's efficiency is two-fold:

1. We compare the computational efficiency of the forward and backward passes of our inverted activation nonlinearity layer in various settings.

2. We demonstrate that the approximation of the nonlinearity gradient does not affect the quality of gradient descent during training.

With these results, we can confidently assert that inverted nonlinearity layers can be safely used in practice to reduce the memory required for training neural networks without any speed or accuracy degradation. All tests and measurements were performed on a single NVIDIA A100 GPU.

### 3.1 COMPUTATIONAL EFFICIENCY

In this section, we compare the computational efficiency of PyTorch Ansel et al. (2024) nonlinearity layers with our Triton-based NVIDIA Corporation implementations of inverted activation nonlinearities. We conduct three groups of tests (with detailed setups presented in appendix A.3):

1. Application of nonlinearity to a tensor.

2. Application of several layers, one of which is a nonlinearity layer. For this purpose, we selected several popular building blocks of modern neural networks:
   - Linear layer + nonlinearity
   - Transformer MLP block: Linear layer + nonlinearity + Linear layer
   - GeGLU Shazeer (2020): Hadamard product of a linear layer + nonlinearity with another linear layer

3. Full neural network training iterations for the following popular network architectures:
   - BERT
   - Llama v3.1 8B

The corresponding measurements are presented in table 1. The results indicate that the difference in computational efficiency between our inverted activation nonlinearity layer and the standard nonlinearity layer within various architectural blocks (such as MLP, Linear + activation, or GeGLU) is

less than 1%. Furthermore, for full-scale neural networks like BERT and Llama, this difference is even more negligible, being less than 0.1%.

|  | Torch (ms) | InvAct (ms) | Precision Bit InvAct (ms) |
|---|---|---|---|
| **Plain GELU** | 0.82 | 0.86 (+5.75%) | 0.86 (+4.96%) |
| **Linear + GELU** | 11.78 | 11.80 (+0.13%) | 11.79 (+0.06%) |
| **Tranformer MLP Block** | 75.41 | 75.46 (+0.06%) | 75.41 (−0.00%) |
| **GeGLU** | 94.08 | 94.13 (+0.05%) | 94.09 (+0.00%) |
| **BERT** | 1112.12 | 1112.62 (+0.05%) | 1112.25 (+0.01%) |
| **Llama3.1-8b** | 182.84 | 182.94 (+0.06%) | 182.80 (−0.02%) |

Table 1: Comparison of computation time (in milliseconds) for different activation function implementatoins and neural network architectures. We compare following architecture blocks: plain nonlinearity layer, fully-connected layer with nonlinearity, Transformer MLP block, GeGLU Shazeer (2020), BERT Devlin et al. (2019) model, Llama3.1-8b model. For each architecture block we compare following activation functions: torch GELU, our inverted activation and precision-bit inverted activation. We show difference in percents w.r.t. torch nonlinearity. Our layer is lower then 1% slower compared to torch nonlinearity inside some computation block, and lower then 0.1% slower inside BERT and Llama.

## 3.2 EFFECT OF THE APPROXIMATION ERROR

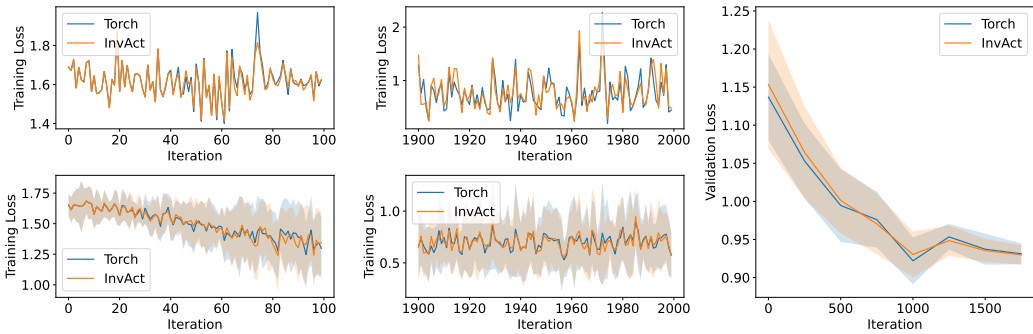

Figure 3: Comparison of BERT model fine-tuning with original and inverted GELU activations. The first row (except the rightmost column) compares two runs with Torch GELU and inverted GELU using identical random seeds. The second row shows losses averaged across 16 runs for each activation function. The leftmost column displays the training loss during the first 100 iterations. The middle column shows the training loss during the last 100 iterations out of a total of 2000. The rightmost column compares the validation losses. The observed losses, both training and validation, are nearly identical, with any discrepancies attributable to minor approximation inaccuracies that are absolutely insignificant and negligible for practical purposes.

In this section, we investigate the impact of our approximation method on the training process and final model performance. To evaluate this, we conducted experiments using two popular neural network architectures: BERT and Llama. We trained these models on standard benchmark tasks using both the original PyTorch GELU activation function and our approximated inverted GELU function for BERT, as well as the PyTorch SiLU activation function and our approximated inverted SiLU function for Llama.

For BERT, we fine-tuned the model on the Yelp Review dataset Yelp (2020) for sentiment classification. We compared the training process of models using both the original GELU activation function and our inverted GELU activation. Comparisons of the training and validation losses are presented in Fig. 3. We observe that the approximation inaccuracy of the inverted activation does not affect BERT's training quality, as both training and validation losses are nearly indistinguishable over all

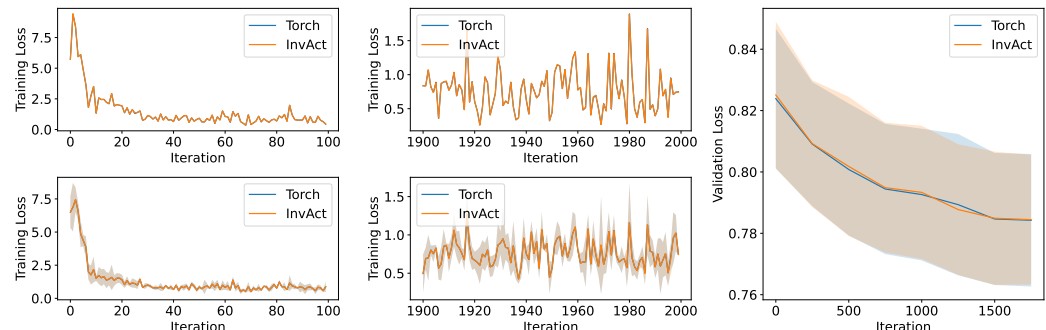

Figure 4: Same as Fig. 3, but for the Llama v3.1 8B model with SiLU nonlinearity. The first row (except the rightmost column) compares two runs with Torch SiLU and inverted SiLU using identical random seeds. The second row shows losses averaged across 16 runs for each activation function. The leftmost column displays the training loss during the first 100 iterations. The middle column shows the training loss during the last 100 iterations out of a total of 2000. The rightmost column compares the validation losses. The observed losses, both training and validation, are nearly identical, with any discrepancies attributable to minor approximation inaccuracies that are absolutely insignificant and negligible for practical purposes.

2000 steps of the training process. The variance produced by different random seeds is significantly larger than any difference between the two activation functions, suggesting that any discrepancies are negligible and on par with machine precision.

We extended this experiment to a significantly larger model, Llama v3.1 8B Dubey et al. (2024), which uses a different activation function – SiLU. We fine-tuned a 4-bit quantized version with LoRA Hu et al. (2022) using ORPO on a combination of DPO datasets[2]. The results for Llama training, presented in Fig. 4, show an even closer match between the original SiLU activation and our inverted SiLU activation compared to the BERT experiment. The training and validation losses are virtually indistinguishable throughout the entire training process. This remarkable similarity further supports our conclusion that the approximation inaccuracy of the inverted activation has a negligible impact on training quality and final model performance.

Based on this comparison of model training using original and inverted activations, we can conclude that the approximation inaccuracy of the InvAct does not affect training quality in any way. Combined with the speed measurements presented in Section 3.1, we can state that the inverted activation nonlinearity layer is a perfect drop-in replacement for GELU and SiLU activation functions.

Furthermore, the seamless integration of our method into existing architectures – without requiring any modifications to pipelines, code, or hyperparameters – highlights its potential for widespread adoption in various deep learning applications. The ability to reduce memory requirements without compromising model accuracy or training dynamics makes our inverted activation approach a promising tool for researchers and practitioners working with memory-intensive models, especially in resource-constrained environments.

### 3.3 COMPARISON WITH FEWBIT

Another method that reduces the memory required by pointwise nonlinearities is FewBit Novikov et al. (2023), a gradient quantization method that saves a quantized version of the nonlinearity gradient. FewBit with one bit per element suffers from a considerable accuracy drop, as demonstrated in Fig. 5. The authors report that using 4-bit quantization, which requires four times more memory compared to our standard InvAct implementation, FewBit does not show any performance degradation. However, even in that setting, our approach is superior: the approximation of $f'(f^{-1}(x))$ is much more accurate than FewBit quantization up to 8 bits! We show $L_2$ and $L_\infty$ approximation errors for FewBit and InvAct in Fig. 5, and our approach demonstrates significantly better approximation quality, even when compared to FewBit using 8 times more bits per element.

---

[2]https://huggingface.co/datasets/mlabonne/orpo-dpo-mix-40k

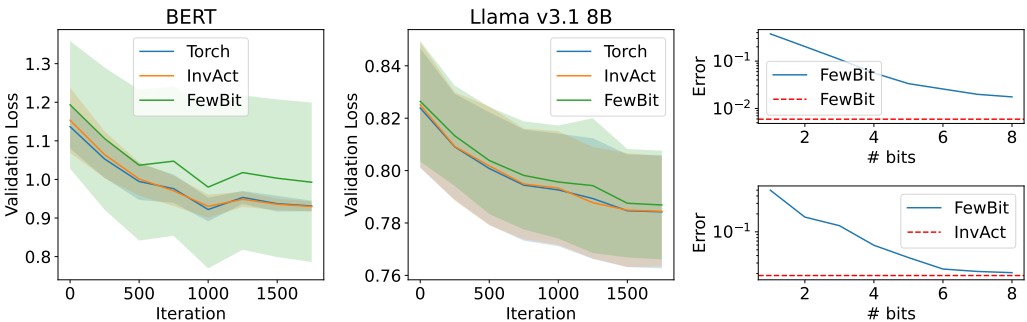

Figure 5: Comparison of InvAct (our) nonlinearity layer with the FewBit Novikov et al. (2023) non-linearity layer. The two left plots show the validation loss for fine-tuning BERT and Llama v3.1 8B, respectively. The BERT model uses the GELU nonlinearity, while the Llama model uses the SiLU nonlinearity. FewBit is used with a 1-bit per element memory budget to match the memory footprint of InvAct. The FewBit approach exhibits significantly worse validation loss, while our InvAct approach is indistinguishable from the standard Torch nonlinearity. The right column shows the $L_2$ approximation error (top plot) and $L_\infty$ approximation error (bottom plot) for FewBit GELU with respect to the number of bits used in gradient quantization (blue line), compared to the approximation quality of InvAct (red dashed line). Our approach always uses only 1 bit per element. InvAct demonstrates superior approximation quality, even when compared to 8-bit FewBit GELU (y-axis is in logarithmic scale).

## 4 RELATED WORK

The efficient scaling of neural, has necessitated innovations in low-precision training, quantization techniques, and various compression methods. These approaches address computational and memory bottlenecks, enabling the training of ever-larger models on hardware-limited resources.

**Quantization methods** aim to compress neural network weights and activations by representing them in lower-precision formats, such as 4-bit integers. One can quantize model weights Jacob et al. (2018); Zafrir et al. (2019); Gholami et al. (2021) or activations that are saved for backward Chen et al. (2021); Novikov et al. (2023). Former are used more in during inference, but approaches to utilize it during training are also exist Liu et al. (2024); Shao et al. (2024); Xu et al. (2024). These methods are especially popular for the fine-tuning stage of training, where reducing memory usage without extensive retraining is crucial. While quantization effectively decreases the memory footprint, it can introduce computational inaccuracies. As a result, these errors lead to degradation in model performance, making the balance between memory savings and maintaining model quality a key consideration when applying quantization.

**Low-precision** training techniques have emerged as another powerful approach to reducing both the memory and computational requirements of large-scale models Micikevicius et al. (2018). One widely adopted format is bfloat16 Kalamkar et al. (2019), which retains a wider range of values compared to traditional FP16 while providing faster computation and reduced memory consumption. More recently work towards even lower precision training in float8 formats have also been explored Wang et al. (2018) for their potential to further reduce memory and computational demands. These formats are particularly attractive for their extreme compression capability, though their usage requires careful consideration to avoid significant accuracy loss. Moreover, widespread adoption of these lower precision formats relies heavily on GPU manufacturers integrating support for them directly into hardware. Until this support is in place, these formats remain experimental and difficult to implement at scale.

**Custom CUDA and Triton implementations** Unsloth (2023); Hsu et al. (2024) have become essential for optimizing critical building blocks in popular architectures, significantly improving memory efficiency and computational speed. For instance, highly fused implementations of MLP blocks Müller et al. (2021) or optimized GeGLU layers are widely used to reduce the overhead of separate operations by merging them into a single, streamlined kernel. Another prominent example

is FlashAttention Dao (2024), which significantly improves the efficiency of attention mechanisms by reducing memory bottlenecks while maintaining performance. These custom kernels, often developed in CUDA or Triton, enable fine-grained optimizations tailored to specific hardware architectures, providing substantial performance improvements. However, the development of such highly optimized components demands significant engineering effort and a deep understanding of both the underlying hardware and neural network operations. Each optimization, whether in matrix multiplications, element-wise operations, or memory management, requires meticulous custom work, and the resulting implementations are typically neural netowrk architecture-specific. Despite the complexity, continuous innovation in these low-level building blocks is critical, as it provides massive improvements in memory usage and training speed, making it possible to push the boundaries of model scaling and performance.

## 5 Limitations

It is important to note that inverted nonlinearities save memory only when the next layer after the nonlinearity also saves its input for the backward pass. Fortunately, this behavior is common in most popular neural network architectures, such as transformers.

Additionally, our method reduces memory usage only for nonlinearity layers. The percentage of memory savings depends on the architecture in use. Furthermore, the presence of complementary techniques may impact these gains. For instance, when training models with checkpointing Griewank & Walther (2000), the memory bottleneck could shift between activations saved during the forward pass and activations recomputed during the backward pass, potentially affecting the benefits of our approach.

Despite these limitations, our highly efficient implementation of inverted activations within the PyTorch framework using Triton kernels makes us confident that our method will see broad adoption. We believe that it will prove invaluable to deep learning practitioners who seek to reduce memory usage without compromising speed order and model accuracy.

## 6 Conclusion

In this work, we presented an innovative method to reduce the memory footprint during neural network training by modifying how activation tensors are saved for backward passes in pointwise nonlinearity layers. By saving the output tensor instead of the input tensor, we significantly decreased the memory requirements for transformer-based architectures such as GPT, BERT, Mistral, and Llama.

Our proposed method includes the development of a space-efficient Boolean indicator to handle the non-invertible nature of these functions, enabling a practical implementation within the PyTorch framework. Additionally, we introduced accurate approximations for the derivatives of the inverse functions of GELU and SiLU, ensuring that training performance remains unaffected while achieving substantial memory savings.

The experiments and approximations demonstrated the feasibility and effectiveness of our approach, with results showing that the training process using our modified nonlinearity layers is indistinguishable from using the original layers in terms of accuracy.

We set out to meet three key criteria for a successful replacement of standard nonlinearity layers: maintaining model speed, preserving model accuracy, and ensuring ease of use. Our proposed method satisfies all three conditions. It does not slow down the neural network, it does not degrade model quality, and it is extremely easy to integrate, making memory savings essentially "free."

We have implemented this method for the PyTorch framework, and the code is available at `https://github.com/removed/for/anonimity`.

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

# A APPENDIX

## A.1 MEMORY TABLE

We perform memory measurements for several popular neural network architectres from Hggging-face model repository: `https://huggingface.co/models`. Memory saving percentage given in a table are calculated as how much less memory is used for stored activations, without taking model size and optimizer statistics in consideration. The following models were used:

- BERT: google-bert/bert-base-uncased
  Measured for sequence length equals 1024
- Audio Spectrogram Transformer: MIT/ast-finetuned-audioset-10-10-0.4593
  Measured for $1024 \times 128$ spectrogram shape
- ViT (Visual Transformer): google/vit-base-patch16-224-in21k
- CLIP: openai/clip-vit-large-patch14
  Measured for text length equals 77 and number of image exaples equal to the number of text examples.

## A.2 APPROXIMATION COEFFICIENTS

Approximations Eqs. (6) to (8) were built by hands. Left part of GELU approximation Eq. (5) were buld with an assstance of PySR Cranmer (2023) framework. Parameters of approximation for GELU:

$$
q^{\text{left}} :
\begin{array}{|c|c|}
\hline
c_0 & +1.6311011311381 \\
\hline
c_1 & +0.16997246666667 \\
\hline
c_2 & -0.06261728 \\
\hline
c_3 & +1.2947087 \\
\hline
c_4 & +1.98055565 \\
\hline
c_5 & +0.22730362 \\
\hline
c_6 & -0.038978495 \\
\hline
c_7 & +1.3295193 \\
\hline
\end{array}
\tag{10}
$$

$$
q^{\text{right}} :
\begin{array}{|c|c|}
\hline
c_0 & -1.383717971214795 \\
\hline
c_1 & +1.558420184350027 \\
\hline
c_2 & +0.044045748018110 \\
\hline
c_3 & +0.032146736769376 \\
\hline
c_4 & -2.119885089843949 \\
\hline
\end{array}
\tag{11}
$$

Parameters of approximation for SiLU:

$$
q^{\text{left}} :
\begin{array}{|c|c|}
\hline
c_0 & -1.310856402130980 \\
\hline
c_1 & +0.848589647031652 \\
\hline
c_2 & -0.162990512595109 \\
\hline
c_3 & +0.002696163985044 \\
\hline
c_4 & -5.770613302664509 \\
\hline
\end{array}
\tag{12}
$$

$$
q^{\text{right}} :
\begin{array}{|c|c|}
\hline
c_0 & +0.217177007595768 \\
\hline
c_1 & -0.507684370508263 \\
\hline
c_2 & +0.079631397669175 \\
\hline
c_3 & +0.357494204859375 \\
\hline
\end{array}
\tag{13}
$$

## A.3 EXPERIMENT SETUPS

- **Plain Nonlinearity**
  Batch size = $2^{25}$.
- **Nonlinearity + Linear**
  Batch size = $2^{15}$, features dimension = $2^{10}$.
- **MLP Block**
  Batch size = $2^{15}$, input/output dimension = $2^{10}$, hidden dimension = $4 * 2^{10}$.
- **GeGLU Block**
  Batch size = $2^{15}$, input dimension = $2^{10}$, output dimension = $4 * 2^{10}$.
- **BERT**
  Batch size = 64, sequence length = 1024
- **Llama v3.1 8B**
  Batch size = 1, sequence length = 512

