# OpenReview forum: "Inverted Activations: Reducing Memory Footprint in Neural Network Training"
_ICLR.cc/2025/Conference — Submitted to ICLR 2025_

### Official Review · Reviewer_r3ne · 2024-10-24

**Soundness:** 2
**Presentation:** 3
**Contribution:** 4
**Rating:** 6
**Confidence:** 4

**Summary:**

This paper derives a method that reduces the amount of data that has to be stored in memory during the forward pass for the backpropagation. By doing so, it is possible to save a significant amount of memory during the training of a neural network. The proposed method is easily implementable and can be deployed in a training procedure with almost no impact
+ on the training duration,
+ on the overall performance of the model,

as it is supported by experimental results.

**Strengths:**

+ The proposed method is derived from a very simple idea, yet it leads to substantial reduction in memory costs, without introducing significant drawbacks or trade-offs (in particular it has no significant impact on either the training time or the model's performance)
+ The method is explained in a very clear way, as well as the research objectives of the paper
+ Thanks to its simplicity, the method is likely implementable within existing frameworks, without changing their core functioning

**Weaknesses:**

The main weeknesses of the paper lie in its experimental section:
+ Some claims of the paper are not supported by experiments, such as line 60: "effectively reducing the memory footprint by nearly $25\\%$ in practice", it would be nice to see empirical evidence of such a claim
+ More generally, no experiment on the memory footprint achieved by the proposed method is shown, yet since the main objective of the paper is apparently to reduce the memory footprint during training, the paper would benefit greatly from such results
+ In section 3.1, the authors use two different frameworks to assess the computational efficiency of their method, but it seems thus difficult to have a clear view on whether the observed gap in computational efficiency is due to the authors method, or to the fact that two different frameworks are being used
+ Finally, could the authors add a theoretical example of how in practice the proposed method would allow to discard some tensors for back propagation computation would make the contribution of the paper more striking

**Questions:**

+ The authors claim that the studied activations are chosen because they are popular choices for transformers, did the authors try to apply their methods to other activations / nonlinearities ?
+ There is a tipo at line 337 ("implementatoins")

---

> ### Author Response · Authors · 2024-11-21
> **Weaknesses: Experiments**
>
> > Some claims of the paper are not supported by experiments.
>
> > More generally, no experiment on the memory footprint achieved by the proposed method is shown.
>
> All claims about memory consumption are based on
> experimental measurements, and specific values can be found in the table in the Introduction section under Listing 1. Memory measurements were conducted as follows:
>
> 1. Instantiate the model.
> 2. Measure the memory consumed by the model parameters.
> 3. Perform a forward pass.
> 4. Measure the currently allocated memory and subtract the model memory to get the memory allocated for activations.
>
> This procedure is then performed for both the standard nonlinearity and the InvAct nonlinearity. The decrease in the memory consumed by activations is reported in the Table in Introduction section.

---

> ### Author Response · Authors · 2024-11-21
> **Weaknesses: two different frameworks**
>
> > In section 3.1, the authors use two different frameworks to assess the computational efficiency of their method.
>
> For all our experiments, we use only one framework: PyTorch. The PyTorch implementation of InvAct is done with Triton kernels, while PyTorch GELU is implemented in CUDA. However, this is a standard practice because Triton is simply a more user-friendly version of CUDA.
>
>  In our time measurements, we carefully ensure that the only difference in the execution of computational blocks (e.g., MLP and GeGLU) lies in the pointwise nonlinearity. All other components are computed identically. Specifically, we use [NVIDIA Nsight Compute](https://developer.nvidia.com/nsight-compute) to verify which kernels are executed on the GPU.

---

> ### Author Response · Authors · 2024-11-21
> **Weaknesses: example of execution**
>
> > Finally, could the authors add a theoretical example of how in practice the proposed method would allow to discard some tensors for back propagation computation would make the contribution of the paper more striking
>
> Consider an example of a Transformer MLP block with two linear layers and a nonlinearity $ f $ between them. For the case of the standard nonlinearity $f = \text{GELU} $, the forward and backward passes would look like this:
>
> Forward pass:
> 1. $ I $ – input of the MLP block.
> 2. $ X = \text{Linear}_1(I) = I W_1 $ – application of the first linear layer. Tensor $I$ is saved for the backward pass.
> 3. $ Y = f(X) $ – application of the GELU nonlinearity. Tensor $X$ is saved for the backward pass.
> 4. $ O = \text{Linear}_2(Y) = Y W_2 $ – application of the second linear layer. Tensor $Y$ is saved for the backward pass.
>
> In total, three tensors are saved for the backward pass: $ I $, $ X $, and $ Y $.
>
> **Backward pass**:
> 1. $ \frac{\partial L}{\partial O} $ – gradient of the loss with respect to tensor $O$, used as input for backward computation.
> 2. $\frac{\partial L}{\partial Y} = \text{Linear}_2\textrm{-backward}\left(\frac{\partial L}{\partial O}\right) = \frac{\partial L}{\partial O} W_2^T$ and $\frac{\partial L}{\partial W_2} = Y^T \frac{\partial L}{\partial Y}$: computation of gradient of loss w.r.t. tensor $Y$ and gradient of second linear layer matrix.
> 3. $ \frac{\partial L}{\partial X} = \frac{\partial L}{\partial Y} \cdot f'(X)$ – backward pass for nonlinearity. Here is the only place where we use saved activation tensor $X$.
> 4. $\frac{\partial L}{\partial I} = \frac{\partial L}{\partial X} W_1^T$ and $\frac{\partial L}{\partial W_1} = I^T \frac{\partial L}{\partial X}$
>
> Now consider the situation with InvAct nonlinearity:
> Forward pass:
> - $I$ – input of the MLP block.
> - $X = \text{Linear}_1(I) = I W_1$ – application of the first linear layer. Tensor $I$ is saved for the backward pass.
> - $Y = f(X)$ – application of the nonlinearity. Instead of saving tensor $X$, a compressed boolean tensor $S = X < \text{const}$ is saved, where $\text{const}$ is a globally defined constant real value. For Precision-bit InvAct, $S$ is saved inside lower precision bit of $Y$ itself.
> - $O = \text{Linear}_2(Y) = Y W_2$ – application of the second linear layer. $Y$ is saved for the backward pass.
>
> In total, we save two tensors ($I$, $Y$), and instead of the full tensor $X$, only the heavily compressed boolean tensor $S$ is saved (if $X$ has data type float 32, then tensor $S$ consumes 32 times less memory). It is  roughly 33% less memory in total.
>
> Backward pass:
> - $\frac{\partial L}{\partial O}$ – gradient of the loss with respect to tensor \( O \), used as input for backward computation.
> - $\frac{\partial L}{\partial Y} = \text{Linear}_2\textrm{-backward}\left(\frac{\partial L}{\partial O}\right) = \frac{\partial L}{\partial O} W_2^T$ and $\frac{\partial L}{\partial W_2} = Y^T \frac{\partial L}{\partial Y}$.
> - $\frac{\partial L}{\partial X} = \frac{\partial L}{\partial Y} \cdot f'\left(f^{-1}(Y, S)\right)$ – here, instead of tensor $X$, which is not saved,we use tensors $Y$ and $S$.
> - $\frac{\partial L}{\partial I} = \frac{\partial L}{\partial X} W_1^T$, and $\frac{\partial L}{\partial W_1} = I^T \frac{\partial L}{\partial X}$.
>
> Comparison: The standard nonlinearity saves three tensors ($I$, $X$, $Y$), whereas InvAct saves two $I$, $Y$, and a compressed boolean tensor $S$. It is  roughly 33% less memory.

---

> ### Author Response · Authors · 2024-11-21
> **Answers to questions**
>
> > The authors claim that the studied activations are chosen because they are popular choices for transformers. Did the authors try to apply their methods to other activations/nonlinearities?**
>
> Most popular computer vision nonlinearities, such as ReLU and Sigmoid, do not require our method because their derivatives can be expressed directly in terms of the output tensors:
>
> $$
> \text{ReLU'}(y) = 0 \text{ if } y = 0 \textrm{ else } 1
> $$
> $$
> \text{Sigmoid'}(y) = y \cdot (1 - y).
> $$
> These are already implemented optimally in PyTorch.
>
> We did not test our approach on other nonlinearities, but there is no fundamental reason why it would not work:
> - Monotonic nonlinearities like SELU require only an appropriate approximation of $f'\left(f^{-1}(y)\right)$ and does not require additional bit per element for inversibility.
> - Unimodal functions can be processed analogously to SiLU and GELU.
> - Nonlinearities with multiple monotonic parts might require more bits per element, potentially reducing the method's applicability.
>
> If there is a specific nonlinearity you would like us to address, we would be happy to explore it.

---

> > ### Comment · Reviewer_r3ne · 2024-11-22
> >
> > I thank the authors for their reply to my concerns. I understand that some of them stemmed from my misunderstanding. I acknowledge this is overall a strong paper, I therefore raise my score to 6.

---

### Official Review · Reviewer_ZY8F · 2024-10-31

**Soundness:** 2
**Presentation:** 3
**Contribution:** 2
**Rating:** 5
**Confidence:** 3

**Summary:**

This article presents a modification to the GELU and SiLU activations to save the output tensor instead of the input, recomputing the input to save approximately a quarter of the memory overhead in practice. They provide several variants of their method, all revolving around the idea of dividing the activation function into monotonous parts which can be inverted. They measure extensively that their approach does not change the computation time and doesn't affect performance.

**Strengths:**

- The method proposed is simple and only requires a drop-in replacement of the activation functions, providing a major memory improvement.
- The proposed inversion methods are novel, notably the approximate inverses of the GELU and SiLU functions.
- The computational efficiency of the method is considered on many different frameworks.
- The paper is clear and well-presented.

**Weaknesses:**

- The main method proposed has very limited novelty outside of the consideration of GELU and SiLU. Other approaches have already proposed inverting the computations of the activation function such as [1] for similar reasons reducing the memory by half and with further benefits.
- At no point comparison with activation/gradient checkpointing is considered, despite the method being extremely similar. Indeed, why not simply store the input tensor only and recompute the output of the activation for efficient gradient computation? This is the same method, but it doesn't require an approximate inversion like proposed here. An extensive literature on checkpointing is necessary to consider here to compare the proposed method to, such as [2-4]. Notably, [5] has a particular consideration for the forward and backward of GeLU. It seems very hard to justify this method compared to more standard checkpointing methods, which computes an optimal trade-off between activation and computation time.
- The approximation error of the derivative of the inverse as represented in Figure 2 shows a non-negligible error for values of $x$ close to 0, which seems a bit worrying.
- Other activation functions are not considered. How is ReLU supposed to work in this framework?

[1] In-Place Activated BatchNorm for Memory-Optimized Training of DNNs, S. R. Bulò et al.

[2] Reducing Activation Recomputation in Large Transformer Models, V. Korthikanti et al.

[3] Efficient rematerialization for deep networks, Kumar et al.

[4] Rockmate: an efficient, fast, automatic and generic tool for re-materialization in pytorch, Zhao et al.

[5] Transcending Runtime-Memory Tradeoffs in Checkpointing by being Fusion Aware, S. Yu et al.

**Questions:**

- How were the coefficients $c_i$ obtained ?
- Why is there an increase in execution time for InvAct only for the Plain GELU? What about the execution time for SiLU?
- The Sign-bit Inverted activation should be explained more clearly.
- Figure 3 requires labels for the rows and columns to be more clear.

**Minor details**

- The abstract in the openreview summary has a formatting error.
- line 182 "based"
- line 701 "assstance"

---

> ### Author Response · Authors · 2024-11-21
> **Answers to questions**
>
> > How were the coefficients obtained?
>
> Coefficients for both parts for SiLU and the right part for GELU were obtained using `scipy.minimize` to minimize the $L_2$ distance between the approximation $q(y)$ and the target function $f^′(f^{−1}(y))$. The symbolic form of these parts was constructed manually. The left part for GELU was constructed and minimized using the PySR framework (https://github.com/MilesCranmer/PySR).
>
> > Why is there an increase in execution time for InvAct only for the plain GELU? What about the execution time for SiLU?
>
> Nonlinearity operations are significantly faster than linear layers, so the minor overhead introduced by InvAct becomes more noticeable when benchmarked in isolation (i.e., for plain GELU) compared to when it is embedded in computational blocks or complete neural network architectures. Additionally, when benchmarking the time for plain nonlinearity (e.g., y = f(x)), both y and x must be calculated and stored in memory, negating the memory-saving benefit.
>
> In contrast, when InvAct is benchmarked inside larger computational blocks, memory savings can lead to more optimal memory access patterns, reducing the relative overhead.
>
> > The Sign-bit Inverted activation should be explained more clearly.
>
> Thank you for pointing that out. We will improve the description of the Sign-bit Inverted activation in the revised manuscript.
> For now, here is more detail: to disambiguate the inverse of the nonlinearity function, we must store an additional boolean value $s = x<T$ along with the nonlinearity output $y = f(x)$. This boolean value requires only one bit of information per element of tensor $x$. The Sign-bit approach uses the lowest precision bit inside $y$ itself.
> $y$, being a floating-point value, has the following representation:
>
> **Image**: [link](https://upload.wikimedia.org/wikipedia/commons/thumb/d/d2/Float_example.svg/1920px-Float_example.svg.png)
>
> The lowest precision bit is bit number 0 in this representation. We can replace its original value with $s$, introducing slight precision error during the forward pass. Sacrificing one precision bit does not significantly impact results, as both float16 and float32 formats provide sufficient precision. Our experiments show no measurable difference in outcomes.
>
> > Figure 3 requires labels for the rows and columns to be clearer.
>
> Thank you for the suggestion. We will improve Figure 3 by adding appropriate labels for the rows and columns.

---

> > ### Author Response · Authors · 2024-11-21
> > **Weaknesses: In-Place Activated BatchNorm**
> >
> > > The main method proposed has very limited novelty outside of the consideration of GELU and SiLU. Other approaches have already proposed inverting the computations of the activation function such as [1] for similar reasons reducing the memory by half and with further benefits.
> >
> > Thank you for pointing out this interesting reference. The method in that paper is limited to easily invertible nonlinearities like LeakyReLU. Moreover, to the best of our knowledge, it does not support cases where $α<0$. Their approach, while efficient for specific use cases, relies on custom fused kernels and cannot generalize neither to other nonlinearities, nor to other computational blocks like Transformer MLP block.
> >
> > In contrast, our method provides a general drop-in replacement for GELU and SiLU, applicable across diverse architectures. It can also handle cases like LeakyReLU with $α<0$, but to the best of our knowledge, PyTorch already does it by saving a boolean indicator tensor $x  < 0$, but maybe not in a bit-efficient way.

---

> > ### Comment · Reviewer_ZY8F · 2024-11-22
> > **> Answers to questions**
> >
> > I thank the authors for answering my questions, which clarified some points for me.
> >
> > Could the authors please answer if the execution time is similar for SiLU to that for GELU?

---

> ### Author Response · Authors · 2024-11-21
> **Weaknesses: other activation functions**
>
> > Other activation functions are not considered. How is ReLU supposed to work in this framework?
>
> The ReLU nonlinearity, $y = ReLU(x)$, does not require additional memory-reduction methods, as its derivative $f′$ can be directly derived from $y$:
> $$
> f^′= 0 \textrm{ if } y = 0 \textrm{ else } 1.
> $$
> Similarly, for example, for sigmoid nonlinearity, $y  =σ(x)$, the derivative can be expressed as $y (1−y)$. For both these nonlinearities PyTorch already does not save input tensor `x`.
> That is why we did not conduct experiments on ResNets, as ReLU is the dominant nonlinearity function in this family of models.
>
> In this work, we focus on the most commonly used nonlinearities in transformer-based architectures: GELU and SiLU. However, our method can be extended to other nonlinearities:
>  * Monotonic nonlinearities like SELU require only an appropriate approximation of $f^′(f^{−1}(y))$ and do not need an additional bit per element.
>  * Any unimodal function can be processed analogically to SiLU and GELU
> * Nonlinearities with multiple monotonic parts may require more bits per element, making the method less applicable.
>
> If there is a specific nonlinearity you would like us to address, we would be happy to explore it.

---

> > ### Comment · Reviewer_ZY8F · 2024-11-22
> > **> Weaknesses: other activation functions**
> >
> > Thank you for clarifying these points. They are interesting and should be included in the final version. This gives a more general view of the method and where it is applicable, as well as the current implementations of common activation functions. However, it also shows that this approach is somewhat limited, as GELU and SiLU are the only major activation functions that require an inverse function.

---

> ### Author Response · Authors · 2024-11-21
> **Weaknesses: Checkpointing**
>
> > At no point comparison with activation/gradient checkpointing is considered
>
> Achieving the same results as our method via checkpointing would require applying it to the pair of layers `Activation + Following Linear Layer`, which has several downsides:
>
> * Performance Overhead: Recomputing GELU during the backward pass adds additional computational cost. Consider, for example, measurements for Transformer MLP Block with standard nonlinearity, InvAct nonlinearity and checkpointed nonlinearity::
>
> **Image**: [link](https://ibb.co/W5HM6wC)
>
> Benchmark shows that checkpointing introduces an overhead of $0.79$%, compared to $0.06$% for our method.
> * Complexity: Checkpointing requires modifications to model architectures and possibly forward computation code, making it less user-friendly. In contrast, our method functions as a drop-in replacement for standard nonlinearities.
>
> Making automatic fusing of such operations and constructing the required frameworks is an intense, ongoing area of research. For example: [link](https://developer.nvidia.com/blog/fusing-epilog-operations-with-matrix-multiplication-using-nvmath-python/).
> Even advanced approaches like [5] (if we understood it correctly) focus only on specific cases of GELU fusing, and they do not generalize to more complex setups involving subsequent linear layers. This limitation arises because PyTorch’s compile backend (which is used in work [5]) does not yet support such intricate fusions. Situation will be even worse if the output of nonlinearity layer is used in several branches of computation: then either such computations all have to be fused separately, further multiplying number of forward-pass GELU recalculations, or not fused at all, which increase number of memory accesses required to perform backward pass. That is not the case for our InvAct nonlinearity approach.
>
> Instead, more widespread applications of checkpointing is when it is used over the entire network. In these cases, our method can be employed in tandem with gradient checkpointing. By reducing the memory footprint of individual nonlinearities, our approach allows methods like [4, 5] find better checkpointing schedules.
>
> In summary: Specialized "small" checkpointing for individual nonlinearities is slower than our method and requires manual implementation, meaning it cannot function as a drop-in replacement. For traditional network-wide checkpointing, our approach can be applied simultaneously, yielding additional memory savings and making larger models feasible.
>
> We will add discussion dedicated to checkpointing to manuscript, thank you!

---

> > ### Comment · Reviewer_ZY8F · 2024-11-22
> > **> Weaknesses: In-Place Activated BatchNorm and Checkpointing**
> >
> > **In-Place Activated BatchNorm**
> >
> > Indeed, the cited work only focuses on the case of (positive) leaky ReLU. However, my point is that this indicates that inverting activations is not particularly a novelty and that this work's main contribution is providing an effective way of inverting GELU and SiLU.
> >
> > **Checkpointing**
> >
> > I find the difference between a computational cost of 0.79% and 0.06% rather minor, but I thank the authors for providing this experiment. Even considering "further multiplying number of forward-pass GELU recalculations", this cost will remain quite negligible, but I agree that inversion is slightly more effective.
> >
> > I do not think that modifying existing architectures to only checkpoint the pre-activations is particularly complicated. Following [this tutorial](https://github.com/prigoyal/pytorch_memonger/blob/master/tutorial/Checkpointing_for_PyTorch_models.ipynb), in particular "Checkpointing any module", it seems like an easy replacement to apply the torch.checkpointing library only for activations. However, it is indeed slightly more involved compared to the drop-in replacement of inversed activations, I agree.
> >
> >
> > **Invertible architectures**
> > Although I did not talk about it in my review, I was wondering about the possible links between this approach and the standard architectures of invertible and reversible networks, such as (i-)RevNets and Reformers. Have the authors considered them?

---

> ### Author Response · Authors · 2024-11-21
> **Weaknesses: The approximation error**
>
> > The approximation error of the derivative of the inverse as represented in Figure 2 shows a non-negligible error for values of $x$ close to 0, which seems a bit worrying.
>
> Please note that in Figure 2, there is a separate $y$-axis on the right of each plot and is scaled by a common multiplier, indicated in the upper-right corner. For example, for GELU, the absolute error has a magnitude of $1e−2$. Our experiments in Sections 3.2 and 3.3 demonstrate that this level of error does not affect training. We also conducted additional experiments using the GELU benchmark with consistent results. Please see our response to Reviewer jQ9m for further details: [link](https://openreview.net/forum?id=Ng1r9kTep4&noteId=4I9fTLXkkw).

---

> > ### Comment · Reviewer_ZY8F · 2024-11-22
> > **> Weaknesses: The approximation error**
> >
> > Indeed, the error value is quite low, and I agree with the authors that the additional experiments provide consistent results. Do the authors have an explanation on why the error particularly seems to spike around $0$?

---

> > > ### Author Response · Authors · 2024-11-22
> > >
> > > At $x \approx−0.75$, the GELU function has a minimum point, and it is where the two monotonic parts meet. In the manuscript, this point is denoted as $T$ (see Figure 1). Due to this transition between the two monotonic regions, the approximation quality is not continuous at TT.
> > >
> > > Additionally, the right-side approximation, $q^{\textrm{right}} (Equation 6 in the manuscript), has a larger error near its left boundary. We do not know any particular reason for that, maybe it's just because the function we are approximating has a sharp descent there: [Image](https://ibb.co/K6tQ8kt).
> > >
> > > The parameters for the approximations presented in the paper were optimized using $L_2$​ error minimization. While this approach ensures overall accuracy, combining $L_2$​ and $L_\infty$ minimization could help make the error distribution more uniform. Since some experiments had already been conducted using the initial variant, we stuck to it at the time of writing the manuscript. However, for a production-ready implementation of our approach, the approximation could be further improved.

---

> ### Author Response · Authors · 2024-11-22
>
> > Could the authors please answer if the execution time is similar for SiLU to that for GELU?
>
> Execution times for SiLU are the same as for GELU. Here are the exact value:
>
> |                            | Torch    | InvAct             |
> |----------------------------|----------|--------------------|
> | **Plain SiLU**             | 0.82     | 0.88 (+7.30%)     |
> | **Linear + SiLU**          | 11.78    | 11.80 (+0.10%)    |
> | **Transformer MLP Block**  | 75.41    | 75.46 (+0.07%)    |
> | **GeGLU**                  | 94.08    | 94.13 (+0.05%)    |
> | **BERT**                   | 1105.50  | 1106.12 (+0.06%)  |
> | **Llama3.1-8b**            | 181.84   | 181.01 (-0.46%)   |
>
> The GeGLU component uses SiLU as a gating unit.

---

> ### Author Response · Authors · 2024-11-26
>
> Thank you for your active feedback. Can you provide some additional comments that we can addresses so that you can consider raising your score?

---

> > ### Comment · Reviewer_ZY8F · 2024-11-27
> >
> > We thank the authors for our discussion. They have clarified some unclear points in the original manuscript, and better explained the relation with other activations functions and their implementation.
> > Still, despite being an easy replacement in transformer architecture, the omnipresence of activation checkpointing, which is more general than the simple use of inverted activations, limits the intended use of inverted activations. (The difference in computation seems too limited to make a true difference in practice).
> > The contributions are also relatively limited, as the use of inverted activation is already widespread, just not to non-monotonic activations. Still, the novel approaches to include the information related to the part of the activation to invert, and the high stability of their proposed inverse function is notable.
> > For these reasons, I am raising my score from a 3 to a 5.

---

### Official Review · Reviewer_jQ9m · 2024-11-09

**Soundness:** 4
**Presentation:** 3
**Contribution:** 3
**Rating:** 8
**Confidence:** 5

**Summary:**

This paper proposes a method, called InvAct, to reduce the memory footprint associated with activation tensors during neural network training by modifying how activation tensors are saved for the backward pass in pointwise nonlinearity layers like GELU and SiLU.
Considering a block of two layers, instead of saving the input tensor for each layer, the authors suggest storing the output tensor of layer 1, which is the input tensor of layer 2. Then during the backward pass the input layer of layer 1 is reconstructed using the inverse of the nonlinearity. Since the nonlinearities considered in the paper do not have straightforward analytical inverses, the paper approximates the inverse functions using simpler functions. The paper shows great deal of memory savings for Transformer based architecture, around 25%, without noticeable impact on accuracy or computational efficiency, and can be used as a drop-in replacement for existing nonlinearity layers in PyTorch.

**Strengths:**

- **Originality**: The approach introduces a novel solution to the memory bottleneck in neural networks training. The proposed solution is straightforward yet impactful adjustment for deep learning workflows.
- **Clarity**: The method is clearly outlined, with pseudocode, detailed experimentation, and well written text.
- **Significance**: The reduction of memory footprint in training aligns well with current trends in model scaling. The method's compatibility with widely-used architectures and frameworks adds further value, especially for resource-constrained training environments.

**Weaknesses:**

- **Experimental Scope**: The evaluation is limited to a few model architectures (BERT and LLama) and tasks (e.g., BERT fine-tuning on Yelp Reviews). Adding results from multiple tasks, such as [GLUE](https://huggingface.co/datasets/nyu-mll/glue), would strengthen claims of general applicability and robustness across varied scenarios.
- **Reporting of Experimental Details**:
Including the fine-tuning details for the results reported in Sections 3.2 and 3.3, perhaps in the appendix could benefit the readers and help future research with reproducing the results reported in the paper.
- **Lack of Task-Specific Performance Metrics**: The results shown in Section 3.2 focus on training/validation loss without reporting task-specific metrics (e.g., accuracy, F1 score for Yelp Reviews). Including such metrics would better contextualize the impact of approximation errors on model quality.
- **Lack of Performance Evaluation for Other Models**: Although memory savings are reported for models like the Audio Spectral Transformer, ViT, and CLIP, no experimental analysis is provided to assess if or how these savings impact model performance. For instance, similar to the BERT and Llama evaluations in Sections 3.1 and 3.2, experiments could show if the method affects performance metrics and how much computational overhead is introduced for these models. Furthermore, in the abstract, the authors mention models like *GPT* and *Mistral* as potential beneficiaries of the method, yet there is no experimental data on memory savings, time overhead, or performance impact for these models. Including analysis or removing these model names would provide a more accurate representation of the paper's scope.

**P.S.**: I am more than willing to raise my score if the concerns mentioned in the **Weakness** and **Questions** section are addressed.

**Questions:**

1. **Sample Size and Standard Deviation**: Are the results in Table 1 averaged over multiple trials? If so, what is the sample size and standard deviation?  If the results show metrics over only a single run, then reporting the results over multiple runs with reports on the standard deviations would better highlight robustness and reproducibility of the method.
2. **Precision Bit InvAct**: I am curious to know if you have performed any performance evaluation as seen in Sections 3.2 and 3.3 on Precision Bit InvAct. The results presented in Table 1 suggest this variant to have less overhead but it is not clear how much error it produces in downstream tasks.
3. **Clarification on Experimental Consistency**: For the FewBit comparison in Figure 5, was the same dataset used as in Section 3.2? If different datasets were used, specifying this in Section 3.3 would enhance clarity.
4. **Sequence Length in BERT Experiment**: In Appendix A.3, the experiment setup for BERT specifies a sequence length of 1024, which exceeds the model’s typical limitation of 512 tokens. Could you clarify how this was achieved? Was a specific technique used to expand the sequence limit to 1024? If so please share more details.
## Minor suggestions:
1. **Additional Details Needed in Listing 1**: In Listing 1, memory savings for the Llama model are not shown, despite Llama being used in Section 3 for performance analysis. Including this information would create a more comprehensive view of the memory savings achievable with this approach.
2. **Sentence Rewrite in Line 464**: In the sentence, "Former are used more in during inference," the word "in" is unnecessary. The corrected sentence should read:
  > "Former are used more during inference."

---

> ### Author Response · Authors · 2024-11-21
> **Answers to questions**
>
> > Sample Size and Standard Deviation in Table 1
>
> Yes, all measurements in Table 1 are averaged across several runs. We ensured enough measurements were taken so that the standard error of the mean is several orders of magnitude lower than the differences between the compared values. To simplify table readability, we decided to omit the error levels. We will add this information to the table caption—thank you for pointing this out!
> Here are the details:
> * Plain GELU, Linear + GELU, and MLP Block: each was averaged across 1000 runs, with standard errors of the mean below 2.5e-4, 1e-3, and 3e-3, respectively.
> * GeGLU: Averaged across 100 runs, with a standard errors below 3e-3.
> * BERT and Llama: Each averaged across 10 runs, with a standard error below 1e-1.
>
>
> > Precision Bit InvAct
>
> The results for Precision Bit InvAct in our experiments were identical to those of the standard InvAct implementation. We attribute this to the fact that full-precision floating-point numbers provide sufficient precision for pointwise nonlinearities, specifically, and neural network training in general. This sufficiency explains why reducing floating-point precision is viable in the first place. Thus, one less precision bit in Precision Bit InvAct does not lead to performance degradation.
>
> As stated in Section 2.1, this may change under more aggressive training settings involving extremely low-precision data types, like float8. For this reason, we concentrated on the standard InvAct implementation and reserved the Precision Bit variant for future, more rigorous testing.
>
>
> > Clarification on Experimental Consistency
>
> Yes, the comparison with the FewBit backward approach in Section 3.3 was performed using the same Llama v3.1 experimental setup as in Section 3.2.
>
>
> > Sequence Length in BERT Experiment
>
> The sequence length used was 512. The value of 1024 in the paper was an error. Thank you very much for noticing this — it has been corrected in the manuscript.

---

> ### Author Response · Authors · 2024-11-21
> **Weaknesses: GLUE Experiments**
>
> In our work, we aim to demonstrate that approximation errors have no measurable impact on final performance metrics. However, achieving this goal involves challenges:
> 1. Statistical significance: Proving the absence of a difference is inherently more difficult than proving its presence. This requires conducting numerous runs for credibility.
> 2. Comprehensive evaluation: There will always be "one more" benchmark or model architecture to test.
>
> To address these challenges, we focused on demonstrating that approximation errors are so minor that training losses closely follow those of the original training process — both for training and validation losses and for task-specific metrics.
>
> Nevertheless, we performed additional experiments on the GLUE benchmark using the RoBERTa-base model (based on the configuration taken from [here](https://github.com/facebookresearch/fairseq/blob/main/examples/roberta/README.glue.md)).
> For each model (standard nonlinearity and InvAct), we conducted 16 runs per GLUE dataset. Below, you can see the corresponding metrics for each sub-dataset:
>
> **Image**: https://ibb.co/pxR4Gdg
>
> **Result**: InvAct matches the standard nonlinearity perfectly.
>
> **Statistical test**: A Mann–Whitney U test on average metric values shows no statistically significant difference (p-value = 0.89).
>
> We also conducted additional experiments on the Recognizing Textual Entailment (RTE) dataset:
> We ran 20 runs of 1-bit FewBit backward and compared it with 20 runs of the standard nonlinearity. The Mann–Whitney U test shows that 1-bit FewBit performs statistically significantly worse than the standard nonlinearity (p-value = 0.027).
> In contrast, InvAct does not exhibit a statistically significant difference. To confirm this, we performed an additional 128 runs for both the standard nonlinearity and InvAct. Even with 128 samples for each (6 times more than for 1-bit FewBit), there was no statistical significance (p-value = 0.7).
> The respective boxplots are provided below:
>
> **Image**: https://ibb.co/MN6zW3d
>
> For completeness, we also provide training losses and task-specific metrics averaged across 16 runs to further validate that InvAct’s minor approximation error do not affect the training process:
>
> **Image**: https://ibb.co/2WgcHxJ
>
> **Image**: https://ibb.co/nsRB1hQ
>
> Moreover, as discussed in Section 3.3, other methods report no loss of quality for 4-bit quantization. We demonstrate that our approximation error is smaller than even 8-bit quantization errors. This further supports our claim that InvAct has no noticeable influence on neural network training, which is inherently noisy, stochastic, and robust to larger perturbations.

---

> ### Author Response · Authors · 2024-11-21
> **Weaknesses: Model Choice and Computational Overhead for other models**
>
> **Model Choice**
>
> Transformer-based models are a dominant architecture type, often using GELU and SiLU instead of plain ReLU. For this reason, we selected BERT as a representative small model and Llama v3.1 8B as a representative large model. We believe these two examples adequately reflect the behavior of other transformer-based models, which would likely show very similar results.
>
> Furthermore, we argue that the results presented in the experimental sections are task-independent. Since the minimal approximation error does not influence the training loss, it should not affect other types of losses and, consequently, task-specific metrics either.
>
>
> **Computational Overhead for Other Models**
>
> We appreciate your suggestion to provide computational overhead analysis for the models mentioned in the Introduction. We will add this analysis for completeness—thank you for highlighting this!
>
> At present, we include computational overhead analysis for fundamental building blocks, such as the Transformer MLP block and GeGLU, in Section 3.1. These results can be easily extrapolated to other models not explicitly analyzed in our work. The computational overhead is consistently less than 0.1%, a level that will be dominated by other network components. This means that, for any full-scale model, the computational overhead is effectively negligible.
>
> The same holds for ViT, CLIP, Mistral, and all other models we tested. Our method introduces no computational overhead on practice, making it a practical and efficient solution for memory savings in transformer-based architectures.

---

> ### Author Response · Authors · 2024-11-27
>
> Thank you very much for the suggested comments. We will correct them in the manuscript. You wrote that your score could be increased if we answer the questions. Is there anything else that we haven't answered or haven't answered fully?

---

> > ### Comment · Reviewer_jQ9m · 2024-11-29
> >
> > Thank you to the authors for their thoughtful and detailed response to my concerns. I greatly appreciate the effort to address the points I raised and the inclusion of additional results that further substantiate the claims made in the paper. The newly added evidence significantly strengthens the paper's contributions and provides greater clarity and robustness to its arguments.
> >
> > This is a well-executed and impactful work that tackles a relevant and challenging problem in the field. The methodology is rigorous, and the results are compelling. I commend the authors for their strong execution and for producing such high-quality research. I am pleased to raise my score to an 8. Great work!

---

### Meta-Review · Area_Chair_hx67 · 2024-12-21

**Metareview:**

The paper introduces, a memory-saving method for neural network training that leverages approximate inverse functions for pointwise nonlinearities like GELU and SiLU. By saving the output tensor instead of the input tensor and reconstructing the input during the backward pass, InvAct reduces memory usage in transformer-based architectures without significant impact on model accuracy or computational efficiency. The method is simple, generalizable to popular frameworks like PyTorch. Reviewers pointed out and as acknowledged in the limitations that the technical novelty is limited and narrow in scope as it applies to only a specific set of activations. Moreover, the interaction and comparisons with existing methods for memory saving like activation checkpointing is not fully explored. Overall the work is incremental and limited in scope and may be more fitted for another venue.

**Additional Comments On Reviewer Discussion:**

Reviewers highlighted the practicality of InvAct.  However, suggestions for broader experimental scope included evaluating additional models, nonlinearities, and comparisons with checkpointing methods. The authors clarified that InvAct incurs less computational cost and is more user-friendly for individual nonlinearities, while complementing network-wide checkpointing. Concerns about approximation errors in GELU and SiLU were addressed with evidence showing negligible impact on model performance. While some reviewers questioned InvAct's novelty, the authors emphasized its unique applicability to non-monotonic nonlinearities and ease of integration as a drop-in replacement. Limitations to GELU and SiLU were noted, but the authors argued these are the most relevant for transformer architectures and suggested potential extensions to other nonlinearities. Additional experiments and responses led to improved reviewer scores, though questions about the method's generalization and scope indicate areas for future refinement.

---

### Decision · Program_Chairs · 2025-01-22

Reject